# Mesenchymal Stromal Cells-Derived Extracellular Vesicles Regulate Dendritic Cell Functions in Dry Eye Disease

**DOI:** 10.3390/cells12010033

**Published:** 2022-12-22

**Authors:** Rongjie Guo, Qi Liang, Yun He, Chenchen Wang, Jiaxuan Jiang, Taige Chen, Di Zhang, Kai Hu

**Affiliations:** Department of Ophthalmology, The Affiliated Drum Tower Hospital, Medical School of Nanjing University, Nanjing 210008, China

**Keywords:** mesenchymal stromal cells-derived extracellular vesicles (MSC-EVs), dry eye disease, dendritic cells, antigen presenting cells, Th17 cells, inflammation

## Abstract

We explored the therapeutic efficacy of Mesenchymal stromal cells-derived extracellular vesicles (MSC-EVs) and its inhibition of the functions of dendritic cells (DCs) in dry eye disease (DED). MSC-EVs were isolated from the culture supernatants of mesenchymal stromal cells (MSCs) and characterized. In vitro, human corneal epithelial cells (HCECs) were cultured in hyperosmotic medium to simulate the DED hyperosmotic environment and treated with MSC-EVs. Cell viability was assessed, and the expression of inflammatory cytokines was quantified. Next, we induced DED in female C57BL/6 mice and divided the mice into groups treated with either MSC-EVs or phosphate buffer solution (PBS) eye drops. Disease severity was assessed; mRNA expression of inflammatory cytokines was analyzed by RT-PCR; and Th17 cells were detected by flow cytometry. Lastly, we evaluated DCs by immunofluorescence and flow cytometric analysis to assess its amounts and maturation. MSC-EVs showed protective effects on HCECs under hyperosmotic stress in vitro, suppressing the expression of inflammatory cytokines. In vivo, mice topically treated with MSC-Evs presented reduced DED disease severity compared to PBS-treated mice. MSC-Evs downregulated the expression of inflammatory cytokines, including TNF-α, IL-6, and IL-1β, as well as the frequency of Th17 cells. Further investigation showed that MSC-EVs suppressed the increase of amounts and the maturation of DCs in DED. Changes of morphological characters of DCs were also inhibited by MSC-EVs. Our study revealed that MSC-EVs suppressed ocular surface inflammation by inhibiting DCs activation-mediated Th17 immune responses, explicating the therapeutic potential of MSC-EVs in DED and other ocular surface diseases.

## 1. Introduction

Dry eye disease (DED) is a highly prevalent ocular disease, with 5–50% of the general population clinically diagnosed with DED [1]. DED is characterized by a loss of tear film homeostasis, accompanied by many ocular symptoms. Multiple factors contribute to the pathogenesis of DED, including tear film instability and tear hyperosmolarity, ocular surface damage and inflammation, etc. [2]. Among these, inflammation is considered a crucial factor in the continuous development of DED, involving innate immunity, adaptive immunity, and other immunological mechanisms which are yet to be elucidated [3,4,5]. Therapeutic methods targeting ocular surface inflammation have emerged as novel strategies for DED treatment [6].

The cornea was endowed with various kinds of antigen presenting cells (APCs), the density of which increases from the center to the limbus [7]. Dendritic cells (DCs) are considered an important kind of APCs in DED immune response, playing a critical role in both innate and adaptive immunity [8]. Under basal conditions, DCs are able to induce and maintain immune tolerance. These DCs are immature DCs, which express less major histocompatibility complex class II (MHC-II) and co-stimulator B7 (CD80 and CD86) [9,10], and are essential for preventing immune damage to healthy tissue [11]. However, immune homeostasis on the ocular surface can be disrupted when tear film instability and tear hyperosmolarity occur [12]. TNF-α, IL-6 and IL-1β produced by epithelial cells promote DCs maturation and upregulate the expression of MHC-II, CD80 and CD86 on DCs [13]. Mature DCs migrate to the draining lymph nodes (dLNs) to initiate adaptive immune response during DED, promoting T cell polarization into Th17 and Th1 [14,15]. Effector T cells then migrate to the cornea and conjunctiva and secrete IFN-γ, IL-17, and other inflammatory cytokines, further aggravating ocular surface damage [16,17]. Thus, as key cells in DED immune response, inhibition of DCs function has a therapeutic effect on DED [18].

Mesenchymal stem cells (MSCs) are stem cells extracted from mesenchymal tissues, possessing the capacity to modulate immune response and promote tissue regeneration [19,20]. Several clinical trials have demonstrated the efficacy of MSCs to treat ocular diseases, making MSC therapy a promising treatment strategy for multiple ocular diseases [21,22,23,24]. However, topical use of MSCs in the eye has also led to many serious complications [25]. Thus, the ocular administration of MSC therapy requires a more cautious evaluation. Immunomodulation and regeneration-promoting effects of MSCs largely depend on the paracrine mechanisms [26], and the use of MSC-derived products may overcome the safety concerns of MSCs. MSC-derived extracellular vesicles (MSC-EVs) are nano-sized vesicles secreted by MSCs, which contains multiple biologically active substance including nucleic acids, proteins (cytokines and chemokines), and lipids. These encapsuled substances mediate the various biological functions of EVs [27]. In view of the structural and functional characteristics of MSC-EVs, MSC-EVs may be considered a novel treatment method for ocular diseases.

A recent study [28] reported that human adipose tissue-derived stem cells-derived extracellular vesicles (hADSC-EVs) inhibited NLRP3 inflammasome activation and reduced disease severity in DED. However, the anti-inflammatory and immunoregulatory effects of MSC-EVs in DED requires further clarification. In this study, we investigated the therapeutic efficacy of MSC-EVs in DED, and demonstrated the role of MSC-EVs in suppressing inflammation by impeding the functions of DCs in DED through in vivo and in vitro experiments.

## 2. Materials and Methods

### 2.1. Cell Culture

The human umbilical cord MSCs (hUC-MSCs) were given by the Clinical Stem Cell Center, The Affiliated Drum Tower Hospital of Nanjing University Medical School. The hUC-MSCs were cultured in DMEM-low glucose medium (Gibco, Grand Island, NY, USA), supplemented with 10% FBS (Gibco), 100 U/mL penicillin, and 100 g/mL streptomycin (Gibco). The 3rd to 7th passages were used for extracellular vesicles isolation.

Human SV40 immortalized corneal epithelial cells (HCECs) (CRL-11135, HCE-2; ATCC, Manassas, VA, USA) were cultured in DMEM/F12 medium (Gibco) with 10% FBS (Gibco) 100 U/mL penicillin, and 100 g/mL streptomycin (Gibco). In DED, due to excessive evaporation of tears, the osmotic pressure in tears becomes higher, and corneal epithelial cells are exposed to the hypertonic environment, leading to cells damage. To simulate the hyperosmotic environment of the ocular surface in DED, HCECs were incubated in serum-free culture medium for 24 h; the culture medium was then changed to hyperosmotic medium, which contains different molality of sodium chloride (NaCl), including 0, 30, 50, 70, 90, and 120 mM, with the corresponding osmotic pressures being 310, 350, 400, 450, 500, and 550 mOsM. The cells cultured in 70 mM were treated with different concentrations of MSC-EVs, including 10 μg/mL, 20 μg/mL, and 50 μg/mL. For the cell viability evaluation, HCECs were cultured in hyperosmotic medium for 12 h and tested by cell counting kit-8 (CCK-8). For RT-PCR, cells were cultured in hyperosmotic medium for 3 h.

### 2.2. Isolation and Identification of hUC-MSC-EVs

When the hUC-MSCs expands to P3-P7, cells were inoculated in a 75 cm^2^ culture flask; when the cell fusion reached approximately 80–90%, the culture medium was changed to serum-free DMEM-low glucose medium for another 48 h of culture. The serum-free supernatants were collected, and differential ultracentrifugation was performed for MSC-EVs isolation. The supernatants were centrifuged for 15 min at 3000× *g* and 1 h at 10,000× *g* to eliminate the cells and large vesicles, then filtered with 0.22 µm filters to discard large vesicles and bacteria. After that, the supernatants were moved to polycarbonate tubes (Beckman Coulter) and centrifuged for 2 h at 100,000× *g* at 4 °C. The pellet was resuspended in 400–600 μL of phosphate buffer solution (PBS) for the following experiments. The morphological features of hUC-MSC-EVs were estimated by transmission electron microscopy (TEM), and nanoparticle tracking analysis (NTA) was performed to detect the size distribution and particle concentration of MSC-EVs. In order to evaluate the protein makers of MSC-EVs, Western blot analysis was performed. EVs were lysed in RIPA lysis buffer (Beyotime, Shanghai, China) for 30 min on ice. The lysates were centrifuged at 13,000 rpm at 4 °C for 20 min, and the supernatants containing the relevant proteins were collected. Each lysate sample contained 60 μg of protein, and was separated by SDS-polyacrylaminde gel electrophoresis (SDS-PAGE) and transferred to polyvinvlidene fluoride (PVDF) membrane (Millipore, MA, USA) for Western blot analyses. The primary antibodies used were: CD9 (abcam, ab236630), CD63 (abcam, ab134045), CD81 (abcam, ab219209), Calnexin (abcam, ab133615). HRP-conjugated anti-rabbit IgG secondary antibodies were used (1:5000, Wanleibio, Shenyang China). The signals were visualized using an ECL assay kit (Vazyme, Nanjing, China).

### 2.3. Animals and DED Mouse Model

All animal experiments were approved by the Institutional Animal Care and Use Committee of Nanjing Drum Tower Hospital and were in accordance with the standards in the Association for Research in Vision and Ophthalmology Statement for the Use of Animals in Ophthalmic and Vision Research. Six–8 weeks-old female C57BL/6 mice were purchased from the Animal Center of Yangzhou University and raised in the Animal Experimental Center of Nanjing Drum Tower Hospital (Nanjing, China). To induce DED, mice were housed in a controlled environment chamber (CEC) (relative humidity: <20%) for 14 days and injected subcutaneously with 0.5 mg/0.1 mL scopolamine hydrobromide (Aladdin, Shanghai, China) four times a day. Mice of the same sex and age were raised in a standard environment and served as normal controls (NC). Three days after DED induction, DED mice were divided into three groups (four mice per group). One group received no treatment and served as DED group. For the other two groups, MSC-EVs (1 μL, 1 μg/μL) or vehicle (PBS, 1 μL) were topically applied four times per day for 11 days to evaluate the therapeutic effects of MSC-EVs. After 14 days of DED induction, mice were sacrificed, and tissue was harvested for subsequent experiments.

### 2.4. Measurement of Tear Production and Corneal Fluorescein Staining

Mice were sedated by intraperitoneal injection of 70 mg/kg ketamine and 10 mg/kg xylazine before measurement of tear production and corneal fluorescein staining. The tear production was measured by phenol red-impregnated cotton threads (Jingming, Tianjin, China) on day 0, 9, 12 and 14. The threads were placed in the lower conjunctival fornix for 3 min per eye, and the length of thread wetted with tears was recorded.

The corneal fluorescein staining was performed for each eye of the mice on day 0, 9, 12 and 14.1 μL of 0.5% fluorescein sodium (Jingming, Tianjin, China) was topically applied to the lateral conjunctival sac. After several blinks, the eyes were rinsed with 0.9% saline and examined by slit-lamp biomicroscope under cobalt blue light. To quantify corneal epithelial damage, corneal fluorescein score (CFS) was assessed in a masked fashion, conforming to the standard National Eye Institute scoring system (0–3 scores for each of the five areas of the cornea).

### 2.5. Single Cell Suspension and Flow Cytometry Analysis

Mice corneas were cut into pieces and digested with 2.5 mg/mL Liberase TM (Sigma-Aldrich, St Louis, MO, USA) in PBS at 37 °C for 35 min and then filtered with 70 μm cell strainers to yield single cell suspensions. Draining lymph nodes were cut into pieces and ground, and the single cell suspension was prepared by filtration through 70 μm cell strainers.

To evaluate CD11c^+^ APCs, single cell suspensions of corneas and draining lymph nodes were stained with PE-CYANINE7-conjugated anti-CD45 (clone: 30-F11, eBioscience, San Diego, CA, USA), FITC-conjugated anti-CD11c (clone: N418, Biolegend, Inc., San Diego, CA, USA), PE-conjugated anti-MHC-II (clone: M5/114.15.2, eBioscience), and APC-conjugated anti-CD86 (clone: GL1, eBioscience) for 30 min at 4 °C. To quantify Th17 cells, single cell suspensions of draining lymph nodes were incubated with cell stimulation cocktail (BD Biosciences, Franklin Lakes, NJ, USA) for 5 h at 37 °C. Surface staining was performed with PE-CYANINE7-conjugated anti-CD45 (eBioscience) and FITC-conjugated anti-CD4 (clone: GK1.5, Biolegend). Following surface staining, cells were resuspended in Fixation/Permeabilization Solution (BD, Franklin Lakes, NJ, USA), and intracellular staining was performed with PE-conjugated anti-IL-17A (clone: eBio17B7, eBioscience). Isotype-matched antibodies were included in all the experiments for controls. Flow cytometry analysis was performed on BD Accuri C6 (BD Biosciences), and the data were analyzed with FlowJo V9.2 (FlowJo, LLC, Ashland, OR, USA).

### 2.6. Immunofluorescence Staining

Immunofluorescence staining was performed as previously described [29]. Briefly, corneas obtained from mice were fixed in 4% paraformaldehyde for 30 min at room temperature (RT). Subsequently, corneas were incubated with 0.5% Triton-X 100 for 30 min at RT to break the cell membrane. Non-specific staining was blocked by incubating the corneas with 5% bovine serum albumin for 1 h at RT. For quantification and morphological analysis, corneas were incubated in antibodies against CD11c (1:200, cat. 117307, Biolegend) overnight at 4 °C. Samples were then washed and mounted with DAPI-containing medium (Abcam, Cambridge, MA, USA). Staining was photographed with a confocal laser scanning microscope (Olympus BX53, Tokyo, Japan) and analyzed using Image J software (NIH, Bethesda, MD, USA). To evaluate the dendritic complexity, two masked observers (QL and YH) evaluated all images and manually counted the dendritic tips of per cell. In cases of more than 10% difference between the two observers, a third observer (KH) also evaluated the images, and the average of these three values was used for the analysis.

### 2.7. RNA Isolation and Quantitative Real-Time PCR

Total RNA was extracted from cultured HCECs, corneas, and conjunctiva tissues using TRIzol reagent (Takara, Kyoto, Japan). The cDNA was compounded by reverse transcription of 1 µg of total RNA using the HiScript II Q Select RT SuperMix (Vazyme, Nanjing, China). RT-PCR was performed on ABI QuantStudio 6 Flex (Invitrogen, Carlsbad, CA, USA) with ChamQ Universal SYBR qPCR Kit (Vazyme, Nanjing, China).

### 2.8. Schiff Periodic Acid Shiff (PAS) Staining

The eyeballs and eyelids collected from mice were fixed in 4% paraformaldehyde, then embedded with paraffin and sectioned into 5 µm thickness. A PAS staining kit (Servicebio, Wuhan, China) was used to stain the sections. All sections were observed and photographed with a digital light microscope (Nikon, Tokyo, Japan).

### 2.9. Statistical Analysis

All in vitro and in vivo experiments were repeated at least three times. GraphPad Prism 8.0 (GraphPad, San Diego, CA, USA) was used for statistical analysis. Data were expressed as mean ± SD. Differences between the two groups were identified using *t*-tests. One-way ANOVA was used to compare the mean values of three or more groups. The statistical significance level was set at *p* < 0.05.

## 3. Results

### 3.1. Identification of hUC-MSCs and MSC-EVs

After differential ultracentrifugation, we obtained 2 mg of proteins of EVs per milliliter. As shown in Figure 1A, the morphology of hUC-MSCs presented as fibroblast-like adherent cells. Transmission electron micrograph demonstrated the saucer-like shape of MSC-EVs (Figure 1B). Nanoparticle tracking analysis determined that the diameter of MSC-EVs ranged from 80 nm to 180 nm (Figure 1C). Exosomes protein markers CD9, CD63, and CD81 were positively expressed in MSC-EVs, as detected by Western blot, while Calnexin was negatively expressed in MSC-EVs (Figure 1D).

### 3.2. MSC-EVs Had Protective Effects on Corneal Epithelial Cells under Hyperosmotic Stress In Vitro

To evaluate the effects of MSC-EVs on HCECs under hyperosmotic stress, we cultured HCECs in hyperosmotic medium containing MSC-EVs. Cell viability data revealed that the increase of the osmotic stress of the medium reduced the cell viability of HCECs (Figure 2A). MSC-EVs at concentrations ranging from 10 to 50 μg/mL protected HCECs from hyperosmotic stress (Figure 2B). Furthermore, MSC-EVs inhibited mRNA expression of inflammatory cytokines TNF-α (*p* = 0.0248), IL-6 (*p* = 0.0005), IL-1β (*p* = 0.0046) in HCECs under 450 mOsM hyperosmotic stress (Figure 2C).

### 3.3. MSC-EVs Reduced Disease Severity in DED In Vivo

To determine the therapeutic effects of MSC-EVs in DED, mice were grouped and received topical application of MSC-EVs, PBS or no treatment during DED induction for 11 days (Figure 3A). After 14 days of DED induction, tear production of DED mice was significantly reduced compared to NC group (1.238 ± 0.2326 mm vs. 4.375 ± 1.157 mm, *p* < 0.0001). Treatment with MSC-EVs significantly increased tear production compared to treatment with PBS (2.663 ± 0.8684 vs. 1.850 ± 0.6047 mm, *p* = 0.0476) and the no treatment DED group (*p* = 0.0005) (Figure 3B). The DED group presented higher CFS scores than NC group (*p* < 0.0001), and topical application of MSC-EVs significantly reduced CFS scores compared with PBS group (*p* = 0.0029) and DED group (*p* < 0.0001), alleviating disease severity (Figure 3C,D). The density of PAS-stained conjunctival goblet cells was also decreased by DED induction (*p* = 0.0044), and MSC-EVs application conjunctival goblet cells density, compared with PBS application (*p* = 0.03) and DED group (*p* = 0.0242) (Figure 3E,F).

### 3.4. Topical Application of MSC-EVs Inhibited Ocular Surface Inflammation in DED

To further investigate whether MSC-EVs can suppress the inflammation in DED, we assessed the mRNA expression levels of inflammatory cytokines in cornea and conjunctiva and the frequency of Th17 cells in dLNs. The RT-PCR results revealed that, compared to the PBS-treated group, MSC-EVs-treated mice had reduced mRNA expression levels of TNF-α (*p*_cornea_ = 0.0137, *p*_conjunctiva_ = 0.001), IL-6 (*p*_cornea_ < 0.0001, *p*_conjunctiva_ = 0.006), and IL-1β (*p*_cornea_ = 0.0042, *p*_conjunctiva_ = 0.2023) in both the conjunctiva and the corneas. More significant differences were observed between MSC-EVs-treated mice and DED mice in the mRNA expression levels of TNF-α (*p*_cornea_ = 0.0088, *p*_conjunctiva_ = 0.0002), IL-6 (*p*_cornea_ = 0.0027, *p*_conjunctiva_ = 0.0001), and IL-1β (*p*_cornea_ = 0.001, *p*_conjunctiva_ = 0.0155) (Figure 4A,B). Flow cytometry data showed that 14 days of DED induction upregulated the frequency of Th17 cells in dLNs compared to NC mice (1.473 ± 0.379 vs. 0.154 ± 0.026 *p* < 0.0001). Topical application of PBS and MSC-EVs reduced the frequency of Th17 cells in dLNs, with MSC-EVs demonstrating more significant effects than PBS (EVs = 0.2633 ± 0.74, PBS = 0.726 ± 0.2699, *p*_DED VS. EVs_ < 0.0001, *p*_PBS VS. EVs_ = 0.0028) (Figure 4C,D).

### 3.5. Topical MSC-EVs Treatment Reduced the Amounts of DCs in DED

Next, we evaluated the effects of MSC-EVs on the amounts of DCs in DED. After topical administration of MSC-EVs, PBS or no treatment, mouse corneas and dLNs were collected. Immunofluorescence staining was performed to determine the density of DCs in the corneas. The amounts of DCs in dLNs were assessed by flow cytometry. Immunofluorescence staining showed that corneas from DED mice had more DCs in both the central cornea (10.68 ± 4.179 vs. 1.07 ± 2.202 cells/mm^2^, *p* = 0.007) and the peripheral cornea (39.84 ± 8.969 vs. 28.61 ± 5.258 cells/mm^2^, *p* = 0.0166) than NC mice. Treatment with MSC-EVs significantly decreased the density of corneal DCs (center = 1.07 ± 2.202 cells/mm^2^, periphery = 18.31 ± 4.974 cells/mm^2^) compared with treatment with PBS (center = 6.676 ± 3.652 cells/mm^2^, *p*_PBS VS. EVs_ = 0.0667, periphery = 31.15 ± 6.570 cells/mm^2^, *p*_PBS VS. EVs_ = 0.0059,) and the DED group (center *p*_DED VS. EVs_ = 0.007, periphery *P*_DED VS. EVs_ = 0.0004) (Figure 5A–D).

The flow cytometry results revealed that the frequency of CD45^+^ CD11c^+^ DCs in dLNs was upregulated in DED (0.64 ± 0.0852 vs. 0.26 ± 0.0636, *p* < 0.0001). Compared with topical PBS treatment, topical MSC-EVs treatment more significantly downregulated the frequency of DCs in dLNs (0.516 ± 0.0635 vs. 0.354 ± 0.219, *p*_PBS VS. EVs_ = 0.0007, *P*_DED VS. EVs_ < 0.0001) (Figure 5E,F).

### 3.6. MSC-EVs Suppressed the Maturation of DCs in DED

Finally, to assess the effects of MSC-EVs on the maturation of corneal DCs, flow cytometry was performed to analyze the expression levels of MHC-II and CD86. Morphological characters of corneal DCs were observed by immunofluorescence staining. Flow cytometry data showed higher frequencies of CD11c^+^ MHC-II^+^ DCs after 14 days of DED induction compared to NC mice (5.603 ± 1.563 vs. 1.857 ± 0.2113, *p* = 0.0147). The inhibitory effects of MSC-EVs (2.717 ± 0.7868, *p* = 0.0461) on the frequency of CD11c^+^ MHC-II^+^ DCs in the corneas of DED mice were more significant compared with PBS (4.133 ± 0.8836, *p*_PBS VS. EVs_ = 0.1068) (Figure 6A,B). Similarly, the frequency of CD11c^+^ CD86^+^ DCs in the corneas of DED mice was also higher than in NC mice (0.9567 ± 0.1266 vs. 0.2533 ± 0.1201, *p* = 0.0022), which was suppressed by topical administration of MSC-EVs (0.2900 ± 0.1411, *p*_DED VS. EVs_ = 0.0037) and PBS (0.7933 ± 0.1419, *p*_PBS VS. EVs_ = 0.0121) (Figure 6C,D).

DCs dendritic complexity is represented by the number of dendritic tips per cell. We manually counted the number of dendritic tips per cell on the immunofluorescence micrographs (Figure 6E) to analyze the differences in DCs dendritic complexity between groups. DCs in the corneas of DED group presented increased dendritic complexity when compared to the NC group (7.037 ± 1.698 vs. 5.355 ± 1.496, *p* = 0.0002). DCs in the corneas of EVs group showed decreased dendritic complexity (5.333 ± 1.234) than in the PBS (6.810 ± 1.914, *p* = 0.0131) and DED groups (7.037 ± 1.698, *p* = 0.0015) (Figure 6E,F).

After 14 days of DED induction, dLNs were collected for flow cytometry analysis. Figure 6G,H shows that the frequency of CD11c^+^ MHC-II^+^ DCs in the dLNs of DED mice was higher than NC mice (2.793 ± 0.6795 vs. 0.88 ± 0.1758, *p* = 0.0092), and treatment with MSC-EVs reduced the frequency of CD11c^+^ MHC-II^+^ DCs (0.9967 ± 0.2501, *p* = 0.0127). MSC-EVs were significantly more effective than PBS (1.59 ± 0.4158, *p* = 0.1016) in suppressing the maturation of DCs. The frequency of CD11c^+^ CD86^+^ DCs in the dLNs of the four groups also showed the same trend as CD11c^+^ MHC-II^+^ DCs in the dLNs (NC = 0.295 ± 0.02517, DED = 0.5980 ± 0.0795, *p*_DED VS. NC_ = 0.0002, PBS = 0.492 ± 0.0955, EVs = 0.335 ± 0.05802, *p*_DED VS. EVs_ = 0.0009, *p*_PBS VS. EVs_ = 0.024) (Figure 6I,J).

## 4. Discussion

In DED, the immune homeostasis of the ocular surface is disrupted, and an inflammatory state is established [30]. Inflammation responses in DED involve multiple immune mechanisms. The corneal epithelium is damaged by desiccation stress and releases inflammatory cytokines, which can promote the maturation of resident APCs [13,31]. Mature APCs capture antigens and migrate to the draining lymph nodes, activating naïve T cells, which subsequently travel to and damage the ocular surface [30]. This vicious inflammatory cycle greatly contributes to the pathogenesis and maintenance of DED [32]. Thus, for the treatment of severe DED, breaking the vicious cycle of DED inflammation is considered the core approach [30]. In this study, we demonstrated that MSC-EVs could inhibit inflammation by intervening in the vicious cycle of DED inflammation in multiple aspects, and hold great promise as a novel treatment for DED (Figure 7).

Desiccation stress induces epitheliopathy, which further leads to tear film instability [33]. Injured epithelial cells express cytokines, chemokines and MHC-II [17,34], release self-antigens, and contribute to subsequent immune responses. Therefore, protecting the epithelium from damage caused by desiccation stress is the key approach in DED treatment strategies. Prior studies have reported that MSC-EVs promote ocular surface tissue regeneration in multiple disease models [28,35,36]. Our in vitro experiments proved that MSC-EVs could play a protective role in a hyperosmotic environment, and the in vivo results confirmed the same effects.

As shown in our in vivo results, healthy corneas harbored a relatively small amount of DCs, which were significantly upregulated after DED induction. It has been demonstrated in multiple diseases that the increase of DCs is related to the inflammatory state of the ocular surface [37,38,39]. To maintain corneal immune privilege, resident DCs express low levels of MHC-II and CD86 [40]. After 14 days of DED induction, we observed increased frequency of MHC-II^+^ and CD86^+^ mature DCs in the corneas and dLNs. The results from our in vivo experiments indicated that topical application of MSC-EVs reduced the amount and maturation of DCs in the corneas and dLNs. Activation and maturation of DCs depend on inflammatory cytokines and chemokines in the ocular surface [41]. Our in vitro and in vivo results both showed that MSC-EVs downregulated the expression of inflammatory cytokines. Thus, we hypothesized that the inhibitory effect of MSC-EVs on DCs is related to its capability of reducing cytokine expression. Reis et al. reported that MSC-EVs attenuate DC maturation and functions in vitro, and microRNA may represent the mechanisms by which MSC-EVs regulate DC function [42]. In our study, DCs were inhibited after topical application of MSC-EVs eye drops, suggesting that MSC-EVs may exert direct effects on DCs. However, specific substances which mediate the regulation of MSC-EVs on DCs in DED still require further investigation.

Inflammatory stimulation leads to morphological changes of DCs, and the changes in extension and the amounts of dendrites are related to the antigen-capture, presentation, and migration functions of DCs [43]. In multiple inflammatory diseases of the ocular surface, alterations of the dendrites of DCs can be observed [44,45,46]. Our results demonstrated that treatment with MSC-EVs can reduce the amounts of dendrites in DED, suggesting that the functions of DCs were inhibited by MSC-EVs.

The polarization of T cells depends on APCs, especially DCs [47]. Regulation of DC will affect the function of T cells. Therapeutic methods regulating Th17 through DCs have been widely studied in many diseases, such as tumor and infectious diseases [48,49]. Our in vivo results revealed that the amount of Th17 cells in the dLNs was reduced in the mice topically treated with MSC-EVs. The reduction of Th17 cells in the dLNs may be mediated by the suppression of DCs. This hypothesis is supported by previous studies which show that CD11c^+^ APC incubated with MSC-EVs present the capacity to dampen down inflammatory T cell responses in vivo and in vitro [42,50].

In this study, we demonstrated the anti-inflammatory and immune-modulating roles of MSC-EVs in DED. MSC-EVs possess multiple advantages over traditional anti-inflammatory eye drops, such as steroid and non-steroidal anti-inflammatory drugs (NSAID). First, it was observed that MSC-EVs play the dual role of anti-inflammation and promotion of regeneration of damaged tissues. Besides, MSC-EVs can be used as a drug delivery system, serving as carriers to deliver other drugs to the ocular surface [51,52]. Second, MSC-EVs also present the characteristics of accumulating in injured tissues in many diseases [53], and may have the same effects on the ocular surface. Nevertheless, more studies are needed to confirm whether the effects of MSC-EVs exist on the ocular surface.

The research on MSC-EVs is still at the preliminary stage, and can be advanced in several directions. First, little research has been performed to evaluate the safety of the topical administration of MSC-EVs on the ocular surface. Second, the optimum dose and treatment intervals to maintain the long-lasting effects of MSC-EVs are yet to be determined. Lastly, a reproducible manufacturing process of MSC-EVs also requires in-depth exploration and innovation. There is still a long way to go for the application of MSC-EVs on a large clinical scale [54].

In summary, our findings revealed that MSC-EVs exerted therapeutic efficacy in DED, both in vivo and in vitro. Furthermore, the data from our experiments demonstrated that MSC-EVs played anti-inflammatory roles in DED by inhibiting the expression of inflammatory cytokines and reducing the frequency of Th17 cells. Lastly, we discovered that MSC-EVs had an immunosuppressive capability on DCs. MSC-EVs played anti-inflammatory roles by suppressing DCs-driven immune response in DED. MSC-EVs hold great promise as a novel treatment method for DED and other ocular surface diseases.

## Figures and Tables

**Figure 1 cells-12-00033-f001:**
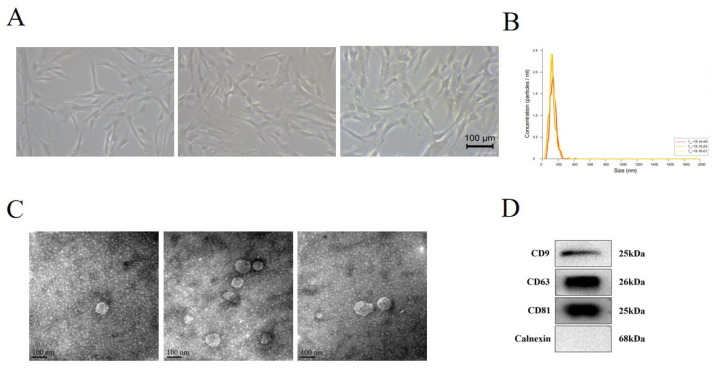
Identification of human umbilical cord MSCs (hUC-MSCs) and MSC-derived extracellular vesicles (MSC-EVs). (**A**) Light microscope image of hUC-MSCs (20×). (**B**) Diameters and concentration of MSC-EVs were detected by nanoparticle trafficking analysis. (**C**) Identification of MSC-EVs by transmission electron micrograph. (**D**) Western blot showed the protein level of MSC-EVs CD9, CD63, CD81, and Calnexin.

**Figure 2 cells-12-00033-f002:**
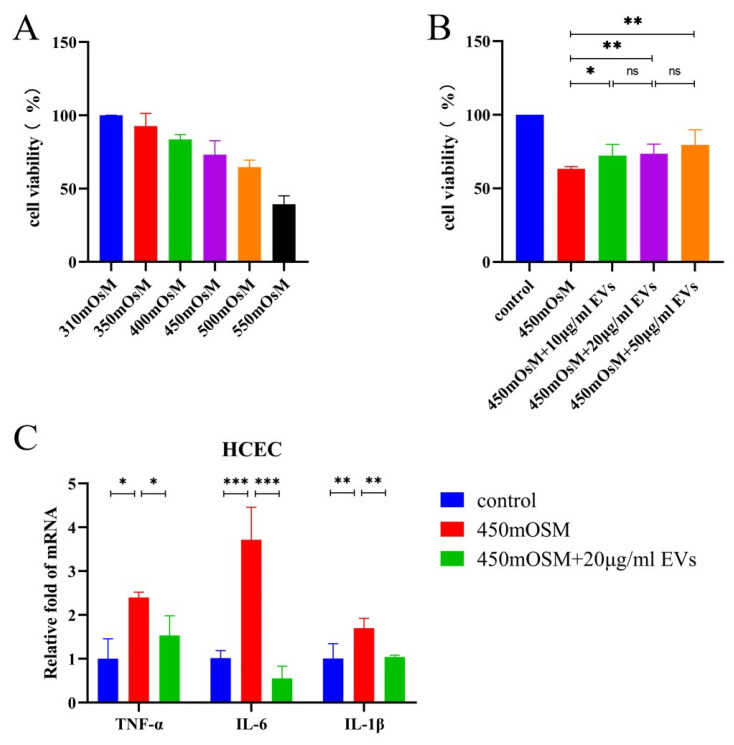
MSC-EVs showed a protective effect on HCECs under hyperosmotic stress. (**A**) Cell viability of HCECs exposed to hyperosmotic medium for 12 h. (**B**) Cell viability of HCECs exposed to hyperosmotic medium and treated with MSC-EVs for 12 h. (**C**) HCECs were incubated in hyperosmotic medium and treated with MSC-EVs for 3 h. The mRNA expression of TNF-α, IL-1β and IL-6 was quantified using RT-PCR. Three independent experiments were pooled. Data are presented as mean ± SD of three independent experiments. (ns: not significant, * *p* < 0.05, ** *p* < 0.01, *** *p* < 0.001).

**Figure 3 cells-12-00033-f003:**
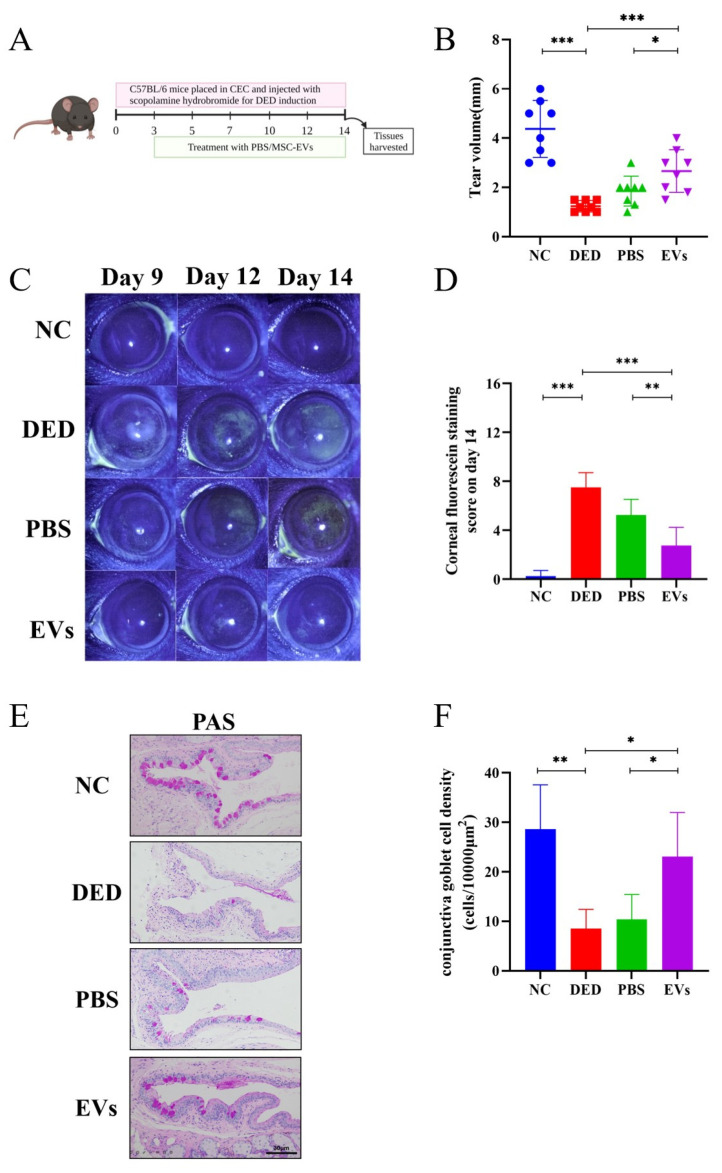
Topical application of MSC-EVs reduced the ocular surface damage of DED mice. MSC-EVs or PBS was topically applied to DED mice to assess the therapeutic effects of MSC-EVs. (**A**) Schedule of DED induction, PBS or MSC-EVs treatment, and tissue harvesting. (**B**) Tear production was measured by phenol red-impregnated cotton threads. (**C**) Representative image of fluorescein sodium staining (2×) shot on day 9, day 12 and day 14. (**D**) Corneal fluorescein score after 14 days of DED induction. (**E**) Representative image (20×) of PAS staining in fornical conjunctiva. (**F**) The density of conjunctival goblet cells was quantified based on PAS staining and location within the conjunctiva. Three independent experiments (n = 8 mice per group) were pooled. Data are presented as mean ± SD of three independent experiments, each consisting of eight mice per group. (ns: not significant, * *p* < 0.05, ** *p* < 0.01, *** *p* < 0.001).

**Figure 4 cells-12-00033-f004:**
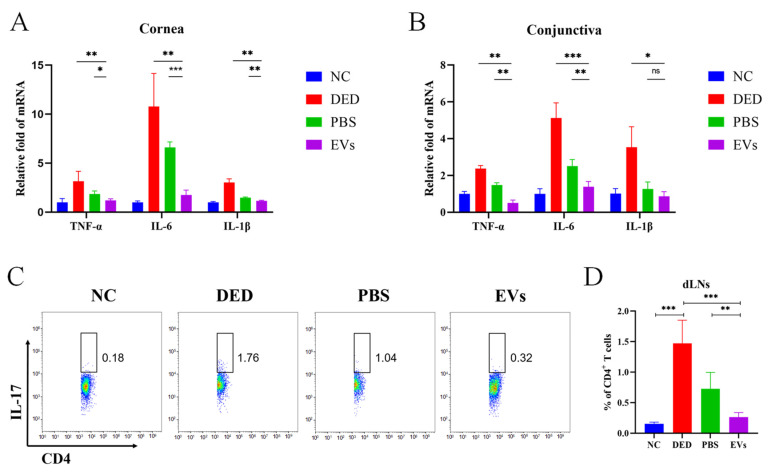
Topical MSC-EVs reduced the level of ocular surface inflammation and protected corneal epithelial cells. We induced DED in a mice model and topically applied MSC-EVs or PBS to assess the inhibitory effects of MSC-EVs on inflammation. Conjunctiva (**A**) and corneas (**B**) of each treatment group were harvested on day 14, and RT-PCR were performed to detect TNF-α, IL-6 and IL-1β mRNA expression. (**C**) Representative flow cytometry plots and bar charts (**D**) showing the frequency of CD4^+^ IL-17^+^ T cells (Th17) in dLNs of mice treated with MSC-EVs or PBS on day 14. Three independent experiments (n = 8 mice per group) were pooled. Data are presented as mean ± SD of three independent experiments, each consisting of eight mice per group. (ns: not significant, * *p* < 0.05, ** *p* < 0.01, *** *p* < 0.001).

**Figure 5 cells-12-00033-f005:**
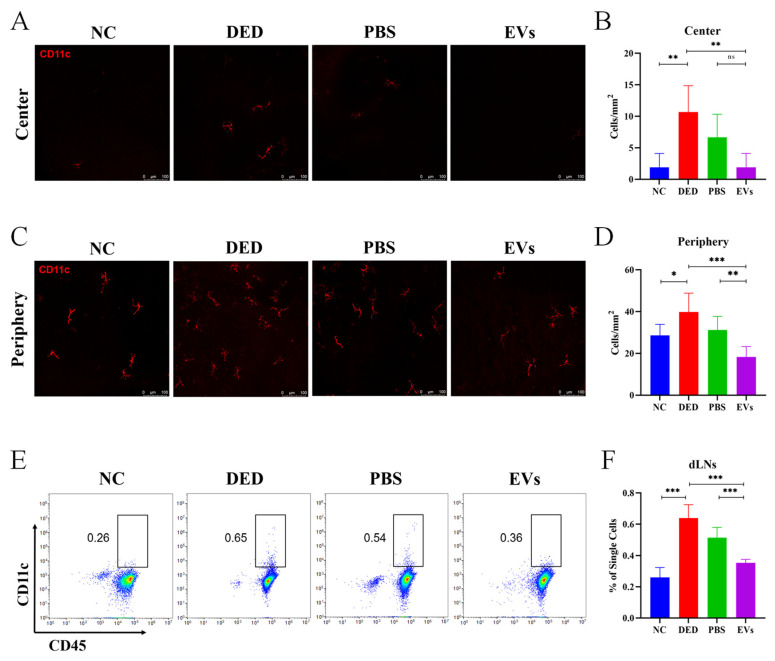
Topical MSC-EVs downregulated the increase in the amount of DCs induced by DED. We assessed the effects of topical application of MSC-EVs on the amount of DCs in mice corneas and dLNs. Representative micrographs (20×) of CD11c (dendritic cells maker)-expressing cells in the central cornea (**A**) and peripheral cornea (**C**) after treatment with topical MSC-EVs or PBS. Bar charts showing the density of DCs in the central cornea (**B**) and peripheral cornea (**D**) of each treatment group. (**E**) Representative flow cytometry plots showing the frequency of DCs in the dLNs of mice treated with topical MSC-EVs or PBS. (**F**) Bar charts showing the frequency of DCs in the dLNs of each treatment group. Three independent experiments (n = 8 mice per group) were pooled. Data are presented as mean ± SD of three independent experiments, each consisting of eight mice per group. (ns: not significant, * *p* < 0.05, ** *p* < 0.01, *** *p* < 0.001).

**Figure 6 cells-12-00033-f006:**
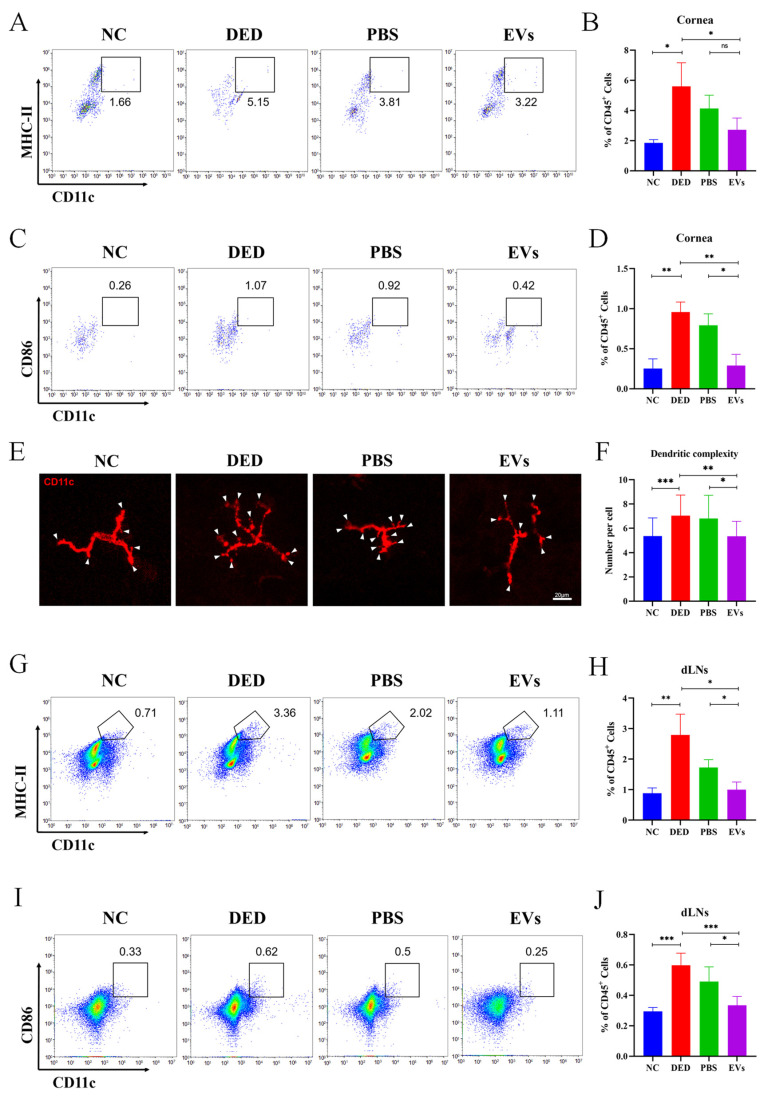
Phenotype change of DCs in DED mice was suppressed by topical application of MSC-EVs. To evaluate the effects of MSC-EVs on the phenotype of DCs, we detected the immunophenotype and morphological characters of DCs. Representative flow cytometry plots showing the frequency of corneal CD11c^+^ MHC-II^+^ DCs (**A**) and CD11C^+^ CD86^+^ DCs (**C**) in mice corneas after treatment with topical MSC-EVs or PBS. Bar charts showing the frequency of corneal CD11c^+^ MHC-II^+^ DCs (**B**) and CD11C^+^ CD86^+^ DCs (**D**) in mice corneas after treatment with topical MSC-EVs or PBS. (**E**) Representative micrographs (20×) of the dendritic complexity (white arrowhead) of DCs in the central and peripheral cornea of each treatment group. (**F**) Bar charts showing the dendritic complexity in the central and peripheral cornea after treatment with topical MSC-EVs or PBS. Representative flow cytometry plots showing the frequency of CD11c^+^ MHC-II^+^ DCs (**G**) and CD11C^+^ CD86^+^ DCs (**I**) in the dLNs of MSC-EVs or PBS treated DED mice on day 14. Bar charts showing the frequency of CD11c^+^ MHC-II^+^ DCs (**H**) and CD11C^+^ CD86^+^ DCs (**J**) in the dLNs of each treatment group. Three independent experiments (n = 8 mice per group) were pooled. Data are presented as mean ± SD of three independent experiments, each consisting of eight mice per group (ns: not significant, * *p* < 0.05, ** *p* < 0.01, *** *p* < 0.001).

**Figure 7 cells-12-00033-f007:**
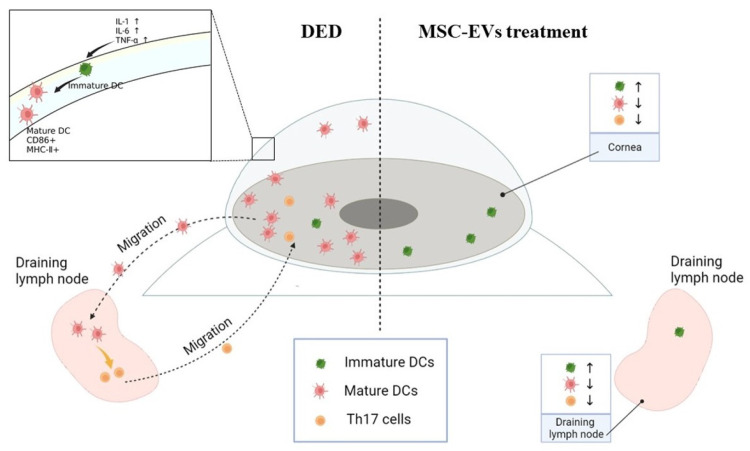
MSC-EVs inhibit ocular surface inflammation by inhibiting DCs activation-mediated Th17 immune responses in DED. Desiccation stress leads to the damage of the epithelium and the release of inflammatory cytokines, such as TNF-α, IL-1β and IL-6, contributing to the proliferation and maturation of DCs. During DED progression, the density and maturation of DCs increase in the cornea. As the initial factors in the adaptive immune response during DED, mature DCs migrate to the regional draining lymph nodes, activating Th17 cells. Th17 cells then travel to the ocular surface, where more inflammatory cytokines are produced and lead to the vicious cycle of DED inflammation. Topical administration of MSC-EVs protected the ocular surface epithelium and inhibited the secretion of cytokines. Proliferation and maturation of DCs were also significantly suppressed by MSC-EVs, which led to further inhibition of the adaptive immune response. Collectively, MSC-EVs could break the vicious cycle of DED inflammation in multiple aspects, and reduce disease severity.

## Data Availability

The data presented in this study are available on request from the corresponding author upon reasonable request.

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
