# Peer review of "Mesenchymal Stromal Cells-Derived Extracellular Vesicles Regulate Dendritic Cell Functions in Dry Eye Disease"

_cells, 2022, doi:10.3390/cells12010033_

Round 1

Reviewer 1 Report

I really enjoyed reading this article about the effect of MSC-derived extracellular vesicles in the regulation of dendritic cell function in dry eye disease. I think this is a nice work that surely interest other scientists in the field. However, there are some comments that must be addressed.

Taking into account the data regarding the size of the EVs isolated obtained during characterization process, these are mainly exosomes. Why don´t the authors refer to this vesicles as such from this moment onwards?

For other scientists not in the field, it would be nice to find in the introduction, a short comment on the mechanism by which the hyperosmolarity induces DED. Also, it will be nice to know how the authors assess the development of the DED in laboratory animals after scopolamine hydrobromide treatment.

The materials and methods section is the most important flaw of this work. I find that this section do not provide enough information and must be extended with further explanations. Here are some of the date I think are missing.

Origin of the hUC-MSCs. The authors do not clarify if the cells were obtained from one of multiple donors or the culture media used for the expansion. Also, the composition of the serum-free media used for isolating the EVs should be stated.

None information whatsoever is given about the amount of protein obtained from the culture media hUC-MSCs (micrograms of protein/number of EVs obtained per ml of culture media), protein preparation method, amount of proteins loaded in the western blot, etc.

This section should be largely expanded giving all necessary details, type of gel used, reference of the antibodies used (this is given by other antibodies used in the work, but not for these ones), etc, etc.

The authors used some acronyms throughout the manuscript without clearly explaining what those are. As an example, in figure 1, they refer to CCK8, a commercial kit to asses cell viability without any explanation of what this is. CFS or PAS staining are other examples.

In figure 1A. Should we assume that 310nm osmolarity is the normal level of osmolarity in a given tissue culture? If so, why not specify this to really know what is used as negative control?

It is not stated in any the graphs is those refer to one single experiment or the average of several experiments.  The “n” should be clearly stated in the figures to be able to assess the soundness of the results. Also, magnification of all pictures should be indicated

Flow cytometry profiles of control (unstained and single stained cells) are not shown.

In Figure 5A it will be nice to have pictures of higher magnification.

Reviewer 2 Report

The manuscript by Guo et al describes the effects of MSC-EVs in a DED mouse model as linked to the inhibitory effects on DC maturation. This is a well-designed and well-presented study, which combines in vitro studies using a human corneal epithelial cell line and in vivo experiments in the DED model. presenting solid evidence for an inhibitory effect of MSC-EVs on the production of inflammatory cytokines in cornea and conjunctivita  and frequency of Th17 cells in DLN. The authors also show a significant decrease in corneal DC maturation following topical application of MSC-EVs in the DED model and propose that the effects on DC maturation are responsible for the therapeutic effects. Although there is no demonstrated direct cause-effect linkage between the effects of MSC-EVs on DC and inflammatory Th17 and cytokines, the presented evidence is highly suggestive for such a link, and the future therapeutic possibilities are quite important.

Author Response

We sincerely thank you for your time and effort in reviewing our manuscript. Your suggestions have enabled us to improve our work.